# OpenReview forum: "Sycophancy Is Not One Thing: Causal Separation of Sycophantic Behaviors in LLMs"
_ICLR.cc/2026/Conference — Submitted to ICLR 2026_

### Official Review · Reviewer_jMZ9 · 2025-10-25

**Soundness:** 3
**Presentation:** 3
**Contribution:** 3
**Rating:** 6
**Confidence:** 2

**Summary:**

The paper argues that “sycophancy” in LLMs is not monolithic. It decomposes the phenomenon into sycophantic agreement (SYA), genuine agreement (GA), and sycophantic praise (SYPR), provides operational definitions, and studies their internal representations. Using DiffMean (difference-in-means) directions on residual stream activations, the authors report that the three behaviors are linearly separable, independently steerable via activation addition, and consistent across model families/scales. They construct controlled synthetic datasets with a knowledge filter (neutral-prompt test) to avoid conflating uncertainty with sycophancy, and show cross-model replications plus an external check on a TruthfulQA sycophancy subset. Overall, the work claims behavior-selective control, suppressing “echoing false user claims” without harming “agreeing when the user is correct.”

**Strengths:**

Clear behavioral formalization. The paper crisply distinguishes SYA, GA, and SYPR and ties them to observable labels with a knowledge filter; Synthetic controls improves reproducibility.

Interpretable method. DiffMean directions on residual activations are lightweight, parameter-free, and easy to inspect; the AUROC and geometry analyses are compelling.

Mechanistic evidence. Steering via activation addition modulates one behavior with minimal cross-effects, strengthening the case for functional separability.

Cross-model consistency. Results replicate across multiple model families/sizes, suggesting the effect is not a single-model artifact.

**Weaknesses:**

The main evidence is on controlled synthetic tasks; only a limited real-world test (TruthfulQA sycophancy subset) is shown. Generalization to open-ended, multi-turn dialogue or socially nuanced settings is unclear.

Deployment considerations not quantified. Steering overhead (per-token, per-layer activation injections), stability under different decoding settings are not measured.

**Questions:**

Do directions learned on one domain/model transfer zero-shot to other domains (e.g., safety, medical advice), languages, or longer conversations? How does selectivity change in truly open-ended dialogue with mixed user intentions?

What is the inference-time cost and latency impact of per-layer steering at α>0? How stable are the effects across temperatures, sampling strategies, and long-context prompts?

---

> ### Author Response · Authors · 2025-11-26
>
> We thank the reviewer for the thoughtful evaluation of our submission and for highlighting the importance of testing beyond controlled, synthetic settings. We address each point in detail below, and we have strengthened the paper accordingly. In particular, we added SYCON-Bench multi-turn conversational results in Section 5 to evaluate generalization in realistic, untemplated dialogue, and we clarified our framing around deployment considerations in Section 5, line 303, noting that our goal is mechanistic understanding rather than production mitigation.
>
> ## Weakness: Evidence is mostly on controlled synthetic tasks; limited real-world testing makes generalization to open-ended, multi-turn, socially nuanced dialogue unclear.
>
> In the revision, we address this directly by adding results on SYCON-Bench, which is intentionally designed to capture the full spectrum of realistic opinion sycophancy. Unlike our controlled datasets, SYCON-Bench is (i) fully open-response, (ii) multi-turn, and (iii) driven by implicit user opinions and presuppositions rather than explicit statements—exactly the aspects of conversational sycophancy that templated datasets cannot capture. This evaluation therefore fills all gaps identified in the reviewer’s comment: it removes templatedness, introduces realistic conversational variability, and probes the exact deployment-relevant behaviors that motivated this work. The results are shown below.
>
> The key metric is **ToF (Turn-of-Flip):** the first turn (0–5) at which the model adopts the user’s false belief.
>   Lower ToF → more sycophantic.
> The baseline ToF is 2.080, meaning the unsteered model typically resists for ~2 turns.
>
> | Behavior Measured | Metric | Baseline | SyA Steer | GA Steer | Selectivity |
> |-------------------|--------|----------|-----------|----------|-------------|
> | **SyA (on-target)** | ToF | 2.080 | −0.260 | −0.020 | **13×** |
> | **SyPr (cross-behavior)** | Praise Rate | 0.00 | 0.00 | 0.00 | — |
>
> The SYCON-Bench results provide clear evidence that our behavioral decomposition holds even in a multi-turn, higher-noise conversational setting. Steering the SyA direction produces a substantial and highly specific effect on sycophancy: it decreases ToF by 0.260 turns, compared to only 0.020 for GA steering—an order-of-magnitude difference. This yields a 13× behavior-specific selectivity ratio, indicating that SyA and GA correspond to distinct latent mechanisms rather than reflections of a single “sycophancy axis.” Cross-behavior leakage is minimal: GA steering has almost no effect on sycophantic collapse, and neither agreement behavior mediates praise, whose rate remains unchanged under all steering directions. Together, these results demonstrate that the three behaviors retain their causal and geometric independence even under significantly more entangled conversational conditions than those in our primary evaluations.
>
> SYCON-Bench brings several strengths as an external validation setting: it tests multi-turn sycophancy; it evaluates a realistic failure mode—the model’s resistance to adopting a user’s false presupposition; and it provides a genuinely naturalistic, untemplated environment for assessing whether our steering directions generalize beyond our construction datasets. Its main limitation is that it only provides sycophancy metrics (SyA) and not GA-specific metrics, meaning that the standard selectivity metric used earlier in the paper cannot be computed in its original symmetric form. So we provide a behavior-specific selectivity comparison—the ratio of the on-target effect (steering SyA) to the off-target effect (steering GA) on the same metric—and the strong separability observed there reinforces the core claims of our paper.
>
>
> ## Weakness: Deployment considerations (overhead, decoding stability) are not quantified
> We intentionally do not evaluate deployment characteristics because the goal of this paper is mechanistic understanding, not proposing a production mitigation. We clarify this more explicitly in the revision (Section 5 Line 303) to avoid suggesting that DiffMean-based steering is intended as a production-ready control method.
>
> Regardless, the computational overhead of DiffMean steering is small. For each forward pass, steering requires only one addition of a single D-dimensional vector where D is the hidden state dimension (D = 1024 for Qwen3-30B residual width).
> This adds roughly 1–2k FLOPs per token (1024 multiplies, once to scale + 1024 adds per token).
>
> By comparison, a single forward pass of Qwen-30B costs on the order of 10⁸–10⁹ FLOPs per token, depending on sequence length and caching. Thus, the steering overhead is <0.00001% of the model’s total compute—negligible relative to the cost of attention and MLP layers. Decoding stability is similarly unaffected: the intervention is a single linear residual addition and does not modify logits, sampling parameters, or attention states beyond the intended directional shift.

---

### Official Review · Reviewer_2GuW · 2025-10-26

**Soundness:** 3
**Presentation:** 3
**Contribution:** 3
**Rating:** 6
**Confidence:** 4

**Summary:**

The paper operationalizes three behaviors on paired user-model turns: (1) sycophantic agreement (SYA), i.e., the model incorrectly agrees with the user's claim, (2) genuine agreement (GA), i.e., the model correctly agrees with the user's claim, and (3) sycophantic praise (SYPR), i.e., exaggerated flattery regardless of the factual content. The authors show that SYA and GA are highly aligned (geometrically) in early layers but diverge in middle layers, while SYPR remains approximately orthogonal. Moreover, activation addition (steering) along learned directions changes the target behavior with minimal cross-effects; this remains true even after subspace removal of other behavior directions. This representational pattern and steerability can be replicated across several instruction-tuned model families and scales.

**Strengths:**

1. The behavioral operationalization is clear, and the knowledge filtering reduces confounds from uncertainty or ignorance.
2. Simple, transparent measures of approaches (DiffMean, cosine geometry, activation addition) lead to consistent and interpretable findings.
3. The subspace removal strengthens the independence claim beyond raw separability.
4. Cross-model replications (Qwen, LLaMA, GPT-OSS), and external test on TruthfulQA SycoEval are insightful.
5. The framing, specifically, precision mitigation of sycophancy without harming GA, is practically valuable.
6. Overall, the work provides valuable practical insights.

**Weaknesses:**

1. Causal claims rest on linear steering.  Have you conducted more direct causal tracing or minimal-intervention experiments?That would strengthen mechanistic conclusions.
2. SYPR measurement depends on a sentiment or praise classifier plus artificial stems. This may inflate orthogonality and selectivity for praise in wider contexts (i.e., selection bias). Have you thought of ways to mitigate such a bias?
3. There is a heavy reliance on single-turn synthetic templates. Multi-turn, socially nuanced dialogs could be further explored.
4. The statistical reporting for steering/selectivity could include CIs and effect-size comparisons across datasets.
5. How do you avoid or mitigate potential lexical or style leakage across datasets? Despite efforts to diversify praise, an explicit lexical-control (e.g., content-word masking or paraphrase-invariance checks) would help.

**Questions:**

See comments above.

---

> ### Author Response · Authors · 2025-11-26
>
> We are grateful to the reviewer for the constructive evaluation. We address each point in detail below, and we have incorporated several clarifications and extensions in response to your feedback. Most importantly, we added SYCON-Bench multi-turn conversational results in Section 5 to assess whether our decomposition holds in realistic, untemplated dialogue settings, and we now report 95% confidence intervals for all steering and selectivity plots (Sections 5 and 6). These additions directly address your concerns regarding causal interpretability, ecological validity, and statistical reporting.
>
>
> ## Weakness: Causal claims rest on linear steering. Have you conducted more direct causal tracing or minimal-intervention experiments?
> We agree that finer-grained causal-tracing methods (e.g., path patching) can offer additional resolution. In this paper, however, our goal is to establish the minimal causal test: if a small, localized linear intervention can selectively modulate SYA, GA, or SYPR—while leaving the others essentially unchanged—then the behaviors must occupy distinct, linearly accessible causal pathways.
>
> Linear steering is therefore an intentional choice: it is the least expressive intervention available. Demonstrating selective control under such a constrained perturbation is strong evidence that the behaviors are truly separable, not artifacts of prompting or correlations.
>
> Additionally, our subspace-removal ablations act as an independent causal check: removing one behavior’s subspace disrupts only that behavior, with the others preserved. This reinforces causal independence without relying on heavier-weight tracing methods.
>
> ## Weakness: SYPR measurement depends on a classifier + artificial stems; could this inflate orthogonality/selectivity (selection bias)?
>
> We agree this is a legitimate concern, and we’ve tried to design the SYPR pipeline so that the classifier + stem play as little structural role as possible.
> - **Classifier + stem are only for measurement, not for learning the directions.**
> The SYPR directions are learned from EOS residuals on our templated datasets, where praise vs. non-praise is defined purely by the construction (including neutral and negative distractors with identical syntactic shapes). The sentiment/praise classifier with the “Assistant: you are …” stem is used only to read out the effect of steering in a controlled surface form, not to define the subspace.
> - **Orthogonality and selectivity do not depend on the classifier.**
> The key orthogonality and separability results (subspace geometry, AUROC discriminability, subspace removal) are all computed directly in activation space, using SYPR vs. non-SYPR labels from the templated datasets—not from the classifier. So any bias in the classifier can affect the magnitude of praise shifts in that particular evaluation setup, but not the core geometric/causal separability claims.
> - **We explicitly try to avoid trivial lexical selection.**
> To mitigate selection bias from the stemmed evaluation, we (a) include neutral and negative continuations with identical stems during classifier training, and (b) use many praise and non-praise templates in the underlying datasets. This forces the SYPR direction to track “praise vs. non-praise” rather than any single phrase.
>
> We agree that this setup still captures a relatively narrow slice of praise and this remains as a limitation. But we believe that we have operationalized this behavior as appropriately as possible for the scope of this paper.

---

> > ### Author Response · Authors · 2025-11-26
> >
> > ## Weakness: Reliance on single-turn synthetic templates; multi-turn, socially nuanced dialogs could be further explored
> >
> > In the revision, we address this directly by adding results on SYCON-Bench, which is intentionally designed to capture the full spectrum of realistic opinion sycophancy. Unlike our controlled datasets, SYCON-Bench is (i) fully open-response, (ii) multi-turn, and (iii) driven by implicit user opinions and presuppositions rather than explicit statements—exactly the aspects of conversational sycophancy that templated datasets cannot capture. This evaluation therefore fills all gaps identified in the reviewer’s comment: it removes templatedness, introduces realistic conversational variability, and probes the exact deployment-relevant behaviors that motivated this work. The results are shown below.
> >
> > The key metric is **ToF (Turn-of-Flip):** the first turn (0–5) at which the model adopts the user’s false belief.
> >   Lower ToF → more sycophantic.
> > The baseline ToF is 2.080, meaning the unsteered model typically resists for ~2 turns.
> >
> > | Behavior Measured | Metric | Baseline | SyA Steer | GA Steer | Selectivity |
> > |-------------------|--------|----------|-----------|----------|-------------|
> > | **SyA (on-target)** | ToF | 2.080 | −0.260 | −0.020 | **13×** |
> > | **SyPr (cross-behavior)** | Praise Rate | 0.00 | 0.00 | 0.00 | — |
> >
> > The SYCON-Bench results provide clear evidence that our behavioral decomposition holds even in a multi-turn, higher-noise conversational setting. Steering the SyA direction produces a substantial and highly specific effect on sycophancy: it decreases ToF by 0.260 turns, compared to only 0.020 for GA steering—an order-of-magnitude difference. This yields a 13× behavior-specific selectivity ratio, indicating that SyA and GA correspond to distinct latent mechanisms rather than reflections of a single “sycophancy axis.” Cross-behavior leakage is minimal: GA steering has almost no effect on sycophantic collapse, and neither agreement behavior mediates praise, whose rate remains unchanged under all steering directions. Together, these results demonstrate that the three behaviors retain their causal and geometric independence even under significantly more entangled conversational conditions than those in our primary evaluations.
> >
> > SYCON-Bench brings several strengths as an external validation setting: it tests multi-turn sycophancy; it evaluates a realistic failure mode—the model’s resistance to adopting a user’s false presupposition; and it provides a genuinely naturalistic, untemplated environment for assessing whether our steering directions generalize beyond our construction datasets. Its main limitation is that it only provides sycophancy metrics (SyA) and not GA-specific metrics, meaning that the standard selectivity metric used earlier in the paper cannot be computed in its original symmetric form. So we provide a behavior-specific selectivity comparison—the ratio of the on-target effect (steering SyA) to the off-target effect (steering GA) on the same metric—and the strong separability observed there reinforces the core claims of our paper.
> >
> > ## Weakness: Statistical reporting for steering/selectivity could include CIs and effect-size comparisons across datasets
> >
> > We include 95% CIs in the steering results graphs (Section 5 Figure 2, Section 5 Figure 3, Section 6 Figure 5) in the revised paper.
> >
> >
> > **Q: How do you avoid or mitigate potential lexical or style leakage across datasets? Would explicit lexical-control tests help?**
> > Thank you for pointing this out. We agree that lexical/style leakage is a potential confound, and we explicitly designed the datasets and evaluation procedures to minimize this risk. Specifically: (1) praise expressions were generated using multiple syntactic structures, polarity-inverted adjectives, and adversarial paraphrases or were not included at all, and (2) we included syntactic praise lookalikes that are not sycophantic (e.g., neutral or negatively valenced adjectives) so that DiffMean cannot rely on simple adjective polarity.

---

### Official Review · Reviewer_ZGmX · 2025-11-01

**Soundness:** 3
**Presentation:** 3
**Contribution:** 4
**Rating:** 8
**Confidence:** 4

**Summary:**

The paper argues that sycophancy in LLMs is three behaviors (sycophantic agreement, genuine agreement, and sycophantic praise). The paper shows that these behaviors correspond to distinct subspaces that diverge in mid layers. Praise is largely orthogonal. Each behavior can be independently amplified/suppressed with minimal side-effects.

**Strengths:**

+ The decomposition is clear and seems well-motivated

+ I like that the authors looked at robustness across datasets and models (shows that the structure generalizes)

+ Writing and figures are very clear

**Weaknesses:**

- There doesn't appear to be any direct cross-dataset steering transfer, that is, directions trained on one template are not shown to steer on disjoint templates

- The selectivity ratio depends on an arbitrary epsilon (and there are no confidence intervals reported); this makes the cross-model comparisons hard to interpret

- The steering setup relies a lot on prompting ("Assistant: You are...") which makes it hard to tell if the results just rely on textual pattersn rather than praise behavior

**Questions:**

- Can the authors evaluate cross-template steering transfer? Learning SYA/GA/SYPR directions on arithmetic and testing steering on CITIES or COMMON CLAIMS to assess behavioral generalization beyond shared geometry?

- How sensitive are the coverage and selectivity results to the knowledge-predicate thresholds?

- Does praise steering persist for free-form praise phrasing beyond “you are ...” and under adjective-family holdouts or paraphrased compliments?

- Are there identifiable attention heads or MLP units that align with the mid-layer SYA–GA split, e.g., circuit level localization

---

> ### Author Response · Authors · 2025-11-26
>
> We appreciate reviewer’s thoughtful assessment of our submission. We address each point in detail below, and we have incorporated the requested clarifications into the revised manuscript. Specifically, we added a new Appendix M presenting a full ε-sensitivity analysis to demonstrate robustness across two orders of magnitude, we added new SYCON-Bench experiments in Section 5 to evaluate whether our behavioral decomposition generalizes to realistic multi-turn conversational settings and we now include 95% confidence intervals in all steering plots in Figures 2,3,5.
>
> ## Weakness: The selectivity ratio depends on an arbitrary epsilon, and no confidence intervals are reported
> We agree that the ε–stabilization term introduces a degree of arbitrariness. Importantly, however, the relative selectivity across behaviors, steering directions, and models is **stable across a wide range of ε values**, and none of our qualitative conclusions depend on its precise choice.
> To make this clear, we conduct a sensitivity analysis (Appendix M) sweeping ε over two orders of magnitude (0.001–0.05). For each ε, we recompute selectivity and normalize by the ε = 0.01 value:
> $$
> \mathrm{NormalizedSel}(\epsilon) = \frac{\mathrm{Sel}(\epsilon)}{\mathrm{Sel}(0.01)}.
> $$
>
> Epsilon-Normalized Selectivity (SyA, Qwen3-30B, Simple Math)
> | α   | ε=0.001 | ε=0.005 | ε=0.01 | ε=0.02 | ε=0.05 |
> |-----|---------|---------|--------|--------|--------|
> | -2  | 10.00×  | 2.00×   | 1.00×  | 0.50×  | 0.20×  |
> | 2   | 10.00×  | 2.00×   | 1.00×  | 0.50×  | 0.20×  |
> | -4  | 8.80×   | 1.87×   | 1.00×  | 0.57×  | 0.26×  |
> | 4   | 9.31×   | 1.92×   | 1.00×  | 0.52×  | 0.22×  |
>
> These results show a consistent multiplicative rescaling across all α values: the selectivity curves follow the *same pattern* regardless of ε. Thus, ε affects only the *numerical scale* of the metric—not the relative distinctions that matter for interpretation. We include this sensitivity analysis in the appendix to make the robustness transparent.
> Thus, the epsilon floor acts only as a numerical stabilizer; it is not responsible for the separability patterns we observe. Our conclusions rely on the geometry of the underlying representations, not on the precise value of the stabilizing constant ε.
>
>
> ## Weakness: Confidence Intervals are not reported for steering
>
> We include 95% CIs in the steering results graphs (Section 5 Figure 2, Section 5 Figure 3, Section 6 Figure 5) in the revised paper.
>
> ## Weakness: The steering setup relies on prompting (“Assistant: you are…”)—could results just reflect textual patterns rather than praise behavior?
>
> We agree this is an important concern, and we designed the setup to avoid that failure mode. The “Assistant: you are …” stem is only used for measurement, not for training or deriving the SYPR directions. The directions themselves come from EOS residual activations across many praise/neutral/negative spans and multiple syntactic templates. We fix the measurement stem purely to remove confounds and make measurement tractable. The prompt is standardized for evaluation clarity, but the learned behavior directions are not tied to any specific phrase pattern.

---

### Official Review · Reviewer_vcc8 · 2025-11-04

**Soundness:** 4
**Presentation:** 3
**Contribution:** 2
**Rating:** 4
**Confidence:** 3

**Summary:**

The paper argues that sycophancy is not monolithic. Using arithmetics on residual activations and subspace geometry across datasets, the authors claim that sycophantic agreement, genuine agreement, and sycophantic praise are linearly separable and causally steerable, with partial replication across model families and a small external test on TruthfulQA-style prompts.

**Strengths:**

1. The paper provides a clear categorization of sycophancy problem by decomposing it into sycophantic agreement, genuine agreement, and sycophantic praise, which helps ground the discussion in measurable behaviors.

2. The setup of using difference-in-means, different evaluation metrics, and activation additions is rigorous and effective in proving the claims. The results replicated across model families and scales, further strengthening the soundness.

3. The results is inspiring and valuable: treating sycophancy as a family of distinct mechanisms rather than a single behavior gives future a better ground to understand the phenomenon.

**Weaknesses:**

1. The datasets are largely synthetic, and the “praise” behavior is constructed in an artificial way that may not capture realistic conversational sycophancy.

2. The methods are largely standard in interpretability research (Mean, linear probing, activation steering) and do not introduce new methodological innovations/ insights beyond their particular behavioral framing.

3. External validation is limited: the only OOD test is small and shows minimal effect, leaving unclear how well the proposed directions generalize to real deployment scenario.

**Questions:**

1. Could the linear directions identified here correspond to mixtures of nonlinear features—e.g., would SAE features reveal similar behavioral boundaries?

2. How do these results compare quantitatively with prior interpretability work in this area?

3. Have you evaluated whether the discovered directions maintain separability when derived from more realistic conversational data?

---

> ### Author Response · Authors · 2025-11-26
>
> We are grateful for the reviewer’s careful evaluation of our work and the valuable comments provided. We address each point in detail below, and we have incorporated two major revisions in direct response to the concerns raised. First, we added new SYCON-Bench experiments in Section 5 to evaluate whether our behavioral decomposition generalizes to realistic multi-turn conversational settings. Second, we revised the Introduction to more clearly articulate the motivation for our controlled dataset design and to highlight the methodological novelty of using linear interpretability tools to separate closely related social behaviors.
>
>
> ## Weakness: Synthetic datasets / artificial praise
> We agree that our praise spans are templated; however, this was an intentional design choice motivated by the specific risks of confounding in this setting. As we discuss in the paper, DiffMean directions are worst-case optimal: they will reliably extract whatever dominant distinctions are present in the data, even if those distinctions arise from subtle spurious correlations or heterogeneous latent behaviors. In tasks like sycophancy—where agreement, politeness, praise, and stance are easily entangled—this can lead to misleading conclusions unless the supervision is extremely controlled. For this reason, our goal was to **atomicize** each behavioral construct and design datasets that minimize ambiguity.
> That said, we fully agree with the reviewer that it is meaningful to test whether these directions hold up in realistic conversational settings. This is why we include the new SYCON-Bench results in the revision (also see table below). SYCON-Bench is explicitly designed to be the opposite of templated: responses are truly open-ended, multi-turn, and involve **implicit opinions and presuppositions**, not explicit ones. This is a significantly harder and more naturalistic scenario than single-turn templated prompts.
> The fact that these directions exhibit selective control in these settings—especially in SYCON-Bench, which is as far from templated as possible—gives us confidence that the behaviors isolated in the controlled datasets correspond to meaningful internal mechanisms rather than template-specific heuristics.
>
> ## Weakness: Methods are standard
> While the underlying tools (DiffMean, activation additions) are indeed very standard, our use of them is novel. To our knowledge, this is the first work to use these methods to separate closely related social behaviors in LLMs—rather than merely detect them. Prior interpretability papers which investigate social behaviors typically treat the phenomenon as a single axis and use linear methods only to measure or reduce it.
>
> The closest prior work is Zhao et al. (2025), which shows that harmfulness and refusal form separable directions. However, that work does not characterize behavioral relationships in the same way that our paper does—using representational geometry or subspace ablations.
>
> We believe this establishes a useful precedent: using simple linear tools to “atomicize” complex social behaviors inside LLMs. The same approach could be applied to behaviors such as **persuasion vs. explanation, deference vs. helpfulness, politeness vs. uncertainty hedging, concession vs. genuine agreement, or moral deference vs. moral reasoning**—all behaviors that are currently treated as single axes but plausibly decompose into distinct internal mechanisms. In our view, the value comes not from inventing new machinery, but from showing that existing interpretability tools can uncover fine-grained, mechanistic distinctions between behaviors previously assumed to be unified.
>
> We highlight this framing more clearly in the introduction.

---

> > ### Author Response · Authors · 2025-11-26
> >
> > ## Weakness: Limited external validation
> > As previously mentioned, in the revision we address this directly by adding results on SYCON-Bench.
> >
> > SYCON-Bench evaluates a form of opinion sycophancy, which models sycophancy in multi-turn dialogue. The key metric is
> > **ToF (Turn-of-Flip):** the first turn (0–5) at which the model adopts the user’s false belief.
> >   Lower ToF → more sycophantic.
> > The baseline ToF is 2.080, meaning the unsteered model typically resists for ~2 turns.
> >
> > | Behavior Measured | Metric | Baseline | SyA Steer | GA Steer | Selectivity |
> > |-------------------|--------|----------|-----------|----------|-------------|
> > | **SyA (on-target)** | ToF | 2.080 | −0.260 | −0.020 | **13×** |
> > | **SyPr (cross-behavior)** | Praise Rate | 0.00 | 0.00 | 0.00 | — |
> >
> > The SYCON-Bench results provide clear evidence that our behavioral decomposition holds even in a **multi-turn, higher-noise conversational setting**. Steering the SyA direction produces a substantial and highly specific effect on sycophancy: it decreases ToF by 0.260 turns, compared to only 0.020 for GA steering—an order-of-magnitude difference. This yields a 13× behavior-specific selectivity ratio, indicating that SyA and GA correspond to distinct latent mechanisms rather than reflections of a single “sycophancy axis.” Cross-behavior leakage is minimal: GA steering has almost no effect on sycophantic collapse, and neither agreement behavior mediates praise, whose rate remains unchanged under all steering directions. Together, these results demonstrate that the three behaviors retain their causal and geometric independence even under significantly more entangled conversational conditions than those in our primary evaluations.
> >
> > SYCON-Bench brings several strengths as an external validation setting: it tests multi-turn sycophancy; it evaluates a realistic failure mode—the model’s resistance to adopting a user’s false presupposition; and it provides a genuinely naturalistic, untemplated environment for assessing whether our steering directions generalize beyond our construction datasets. Its main limitation is that it only provides sycophancy metrics (SyA) and not GA-specific metrics, meaning that the standard selectivity metric used earlier in the paper cannot be computed in its original symmetric form. So we provide a behavior-specific selectivity comparison—the ratio of the on-target effect (steering SyA) to the off-target effect (steering GA) on the same metric—and the strong separability observed there reinforces the core claims of our paper.
> >
> >
> > **Q: Could the linear directions be mixtures of nonlinear features? Would SAEs reveal similar boundaries?**
> >
> > Yes—nonlinear features almost certainly could be found. But our work is grounded in the linear representation hypothesis, which has repeatedly proven sufficient for identifying behavior-relevant directions in LLMs– including for sycophancy in the past. More complex nonlinear representations often look cleaner correlationally but do not translate into stable causal interventions. Linear directions, by contrast, consistently support selective steering, subspace removal, and cross-model generalization—which is exactly what we demonstrate here.
> >
> > Regardless, our goal is an existence proof: to show that these behaviors can be encoded in directions that are linearly accessible and causally steerable. Which linear features were sufficient for.

---

> ### Author Response · Authors · 2025-11-26
>
> **Q: How do these results compare quantitatively with prior interpretability work in this area?**
> A direct quantitative comparison is not feasible because prior work does **not** separate the behaviors we study. Most existing papers—for example, Chen et al. (2025), Rimsky et al. (2024)—treat *sycophancy* as a single construct, mixing opinion agreement, praise, and general deference into one axis. Since these works do not isolate **sycophantic agreement**, **genuine agreement**, or **praise**, their reported numbers are not measuring the same behavioral units analyzed in our paper.
>
> Among the few works that focus on a single subtype—e.g., Wang et al. (2025) on opinion-based sycophancy—we observe similar mechanistic patterns: mid-layer emergence and linear steerability. Our contribution extends this by showing that other subtypes follow *different* geometric and causal trajectories, and that all three behaviors can be *independently steered* with minimal cross-effects.
>
> For similar reasons, mitigation-strength comparisons are not meaningful: our datasets use tightly controlled, known-answer settings designed to isolate specific behaviors, while prior mitigation work evaluates mixed sycophancy subtypes using a single intervention on open-ended prompts.
>
> In short, the novelty of this paper is **not** in outperforming prior work on a shared metric, but in showing that the behaviors those papers treat as one axis are in fact **mechanistically distinct, geometrically separable, and independently controllable**—a dimension of sycophancy that prior interpretability work has not evaluated.
>
> **References**
> - **Zhao et al. (2025).** *LLMs Encode Harmfulness and Refusal Separately.* arXiv:2507.11878.
>   https://arxiv.org/abs/2507.11878
> - **Rimsky et al. (2024).** *Steering Llama 2 via Contrastive Activation Addition.* ACL 2024.
>   https://aclanthology.org/2024.acl-long.828/
> - **Chen et al. (2025).** *Persona Vectors: Monitoring and Controlling Character Traits in Language Models.* arXiv:2507.21509.
>   https://arxiv.org/abs/2507.21509
> - **Wang et al. (2025).** *When Truth Is Overridden: Uncovering the Internal Origins of Sycophancy in Large Language Models.* arXiv:2508.02087.
>   https://arxiv.org/abs/2508.02087

---

### Author Response · Authors · 2025-11-26

We thank all reviewers for their thoughtful and constructive feedback. We have responded to each review separately and believe the revision fully addresses the concerns raised. But please let us know if there are any other questions or concerns!

Below we summarize the main additions and clarifications made in direct response to reviewer comments:

- **Realistic, multi-turn evaluation (Section 5):**
We added a new evaluation on SYCON-Bench, which provides fully open-response, multi-turn, naturalistic dialogues. These results demonstrate that the SyA/GA/SyPr decomposition—and the selectivity of the associated directions—persists in realistic conversational settings.
- **ε-sensitivity analysis (Appendix M):**
In response to concerns about the ε-stabilization term in the selectivity metric, we added a full two-order-of-magnitude sweep showing that all relative selectivity patterns are robust to the choice of ε.
- **Statistical reporting (Figures 2, 3, 5):**
We now report 95% confidence intervals for the steering plots, addressing concerns about uncertainty quantification.
- **Clarified motivation & methodological novelty (Introduction):**
We expanded the introduction to more clearly explain why controlled datasets are required to avoid confounding between closely related behaviors, and to highlight the novelty of using simple linear tools to separate sycophantic sub-behaviors rather than treat them as a single axis.
- **Deployment framing (Section 5):**
We clarified that our goal is mechanistic understanding, not proposing a production mitigation.

We appreciate the reviewers’ time and insight, which substantially improved the clarity and rigor of the paper. Please let us know if any additional questions remain.

---

### Meta-Review · Area_Chair_DELo · 2026-01-06

**Summary:**

All reviewers agree that the paper offers a clear behavioral framing of sycophancy by decomposing it into three categories--sycophantic agreement (SyA), genuine agreement (GA), and sycophantic praise (SyPr)--and using standard linear interpretability tools (difference-in-means directions, cosine geometry, activation steering) to study their representations and steerability in LLMs. They also acknowledge that the results are internally consistent and replicated across several model families and sizes.

However, several key concerns drive my recommendation for rejection:

- **Limited novelty of methodology.** The core techniques, difference-in-means directions on residual activations, linear probing, and activation addition, are well-established and widely used in interpretability work. The paper does not introduce new modeling, optimization, or interpretability machinery; instead, its primary contribution is a specific behavioral decomposition and application of existing tools. Reviewers consistently flag that the methodological contribution is incremental, not conceptually new.

- **Narrow behavioral scope.** The analysis is restricted to only three behaviors (SyA, GA, SyPr). While this decomposition is interesting, it covers a relatively narrow slice of the broader space of sycophantic and socially nuanced behaviors (e.g., deference vs. helpfulness, persuasion vs. explanation, moral deference vs. moral reasoning). As several reviewers note, it is therefore unclear how far these conclusions generalize beyond this particular triad.

- **Heavy reliance on synthetic, templated datasets.** Much of the core evidence is derived from carefully constructed synthetic tasks with templated prompts and artificially defined praise spans. This raises concerns about ecological validity and the possibility that the directions and separability reflect dataset design choices (templates, lexical patterns, classifier decisions) rather than robust mechanisms in real-world dialogue.

- **Limited external / realistic validation.** Although there is some external testing (TruthfulQA sycophancy subset and, in the revision, SYCON-Bench), the bulk of the results still come from controlled single-turn synthetic datasets. Reviewers remain unsure how well the observed linear structure and steering behavior extend to messy, multi-turn, mixed-intent interactions that are more representative of deployment scenarios.

- **Reproducibility and transparency.** Importantly, the synthetic datasets used to construct and evaluate the behaviors, as well as the code implementing the pipelines, are not submitted or made available with the paper. Because a large part of the contribution hinges on the precise operationalization of sycophantic agreement, genuine agreement, and praise, via templates, knowledge filters, and classifiers, this lack of released data and code significantly undermines reproducibility and the community’s ability to verify and build upon the work.

Taken together, incremental methodological novelty, narrow behavioral scope, synthetic-heavy evaluation, and missing datasets/code, these concerns lead me to a negative recommendation.

**Reviewer Concerns:**

The rebuttal and revision do usefully address several technical and evaluation concerns:

- The authors add SYCON-Bench multi-turn experiments, showing that steering SyA vs. GA has differential effects on a realistic “turn-of-flip” metric in naturalistic dialogue, which partially addresses worries about purely synthetic, single-turn settings.

- They provide an ε-sensitivity analysis for the selectivity metric and now report 95% confidence intervals for steering curves, improving statistical transparency.

- They further explain the role of the praise classifier and fixed stems, emphasizing that these are used only for measurement, not for learning the directions, and describe steps taken to diversify templates and mitigate trivial lexical cues.

- They clarify their intended scope (mechanistic understanding rather than production mitigation) and explicitly position their contribution as using simple linear tools to “atomicize” social behaviors.

Despite these improvements, several substantial issues remain unresolved:

- **Novelty remains limited.** The rebuttal reframes the contribution as a novel application of simple linear tools rather than a methodological innovation. While this framing is honest, it does not change the underlying fact that difference-in-means directions, linear geometry, and activation steering have already been widely used to study model behaviors. The paper’s new element is the specific three-way behavioral labeling, which, while useful, is not enough to offset the incremental nature of the techniques.

- **Behavioral scope remains narrow.** The work still focuses exclusively on SyA, GA, and SyPr. The rebuttal argues that the same approach could be applied to other behavior families in future work, but the current submission does not demonstrate such generalization.

- **Synthetic-data dependence and ecological breadth.** Even with the added SYCON-Bench results, the central geometric and steering analyses are still anchored in bespoke synthetic datasets. The real-world evaluations are comparatively shallow and confined to a small set of opinion-sycophancy prompts. Questions about how robust these decompositions are across broader domains, task types, and socially nuanced contexts remain largely open.

- **Reproducibility (my concern, but not raised by the reviewers): missing datasets and code.** Critically, the paper still does not provide the synthetic datasets or code used to define, train, and evaluate these behaviors. Given that the key claims rest on subtle properties of dataset construction (e.g., knowledge filtering, praise templates, neutral/negative distractors), not releasing these resources makes independent verification and extension very difficult. This is a particularly serious limitation for a paper whose main contribution is empirical and analytic, rather than algorithmic.

**Reviewer Scores:**

The likely post-discussion picture is one strong supporter (8), two weak/borderline supporters (6 and 6), and one marginally negative reviewer (4). Given the limited methodological novelty, narrow behavioral scope, dependence on unreleased synthetic datasets and code, and modest real-world reach, I align with the more cautious side of this distribution and recommend rejection.

---

### Decision · Program_Chairs · 2026-01-26

Reject